

# Comparing neural sentence encoders for topic segmentation across domains: not your typical text similarity task

Iacopo Ghinassi[1], Lin Wang[1], Chris Newell[2] and Matthew Purver[1,3]

[1] School of Electronic Engineering and Computer Science, Queen Mary University of London, London, United Kingdom
[2] BBC R&D, London, United Kingdom
[3] Jožef Stefan Institute, Ljubljana, Slovenia

## ABSTRACT

Neural sentence encoders (NSE) are effective in many NLP tasks, including topic segmentation. However, no systematic comparison of their performance in topic segmentation has been performed. Here, we present such a comparison, using supervised and unsupervised segmentation models based on NSEs. We first compare results with baselines, showing that the use of NSEs does often provide improvements, except for specific domains such as news shows. We then compare over three different datasets a range of existing NSEs and a new NSE based on *ad hoc* pre-training strategy. We show that existing literature documenting general performance gains of NSEs does not always conform to the results obtained by the same NSEs in topic segmentation. If Transformers-based encoders do improve over previous approaches, fine-tuning in sentence similarity tasks or even on the same topic segmentation task we aim to solve does not always equate to better performance, as results vary across method being used and domains of application. We aim to explain this phenomenon and the relative poor performance of NSEs in news shows by considering how well different NSEs encode the underlying lexical cohesion of same-topic segments; to do so, we introduce a new metric, ARP. The results from this study suggest that good topic segmentation results do not always rely on good cohesion modelling on behalf of the segmenter and that is dependent upon what kind of text we are trying to segment. Also, it appears evident that traditional sentence encoders fail to create topically cohesive clusters of segments when used on conversational data. Overall, this work advances our understanding of the use of NSEs in topic segmentation and of the general factors determining the success (or failure) of a topic segmentation system. The new proposed metric can quantify the lexical cohesion of a multi-topic document under different sentence encoders and, as such, might have many different uses in future research, some of which we suggest in our conclusions.

Corresponding author
Iacopo Ghinassi,
i.ghinassi@qmul.ac.uk

## INTRODUCTION

Topic segmentation—automatically segmenting a long text into topically coherent units—is a well known problem in natural language processing, and the first step for a number of downstream applications. A long transcript from a news show, for example, could be divided into single news stories so as to help an end user in retrieving more relevant and specific information (*Reynar, 1999*) or a long article could be divided into subsections to aid its reading (*Hearst, 1997*). Whatever the domain of application, topic segmentation is therefore a useful task and it has the advantage of solid theory in discourse structure to draw from, as the sharing of a common topic (usually defined as a collection of cohesive lexical expressions) is among the things that define the local coherence of a written text or a spoken dialogue act (*Halliday & Hasan, 1976*). Because of this, many attempts have been made in order to exploit various indicators of lexical cohesion for segmentation.

In the mean time, however, methods for determining and expressing similarities between texts have seen a major shift in NLP from word count based methods to geometric distance between dense text representations from pre-trained neural representations (*Smith, 2017*), which (as explained later in more detail) we group under the umbrella term of Neural Sentence Encoders (NSEs). Very recently these advances in text representation techniques have been successfully applied to topic segmentation in various unsupervised and supervised settings (*Ghinassi, 2021*; *Harrando & Troncy, 2021*; *Solbiati et al., 2021*; *Lo et al., 2021*; *Lukasik et al., 2020a*). Drawing from these initial attempts, we explore in more detail the use of these recent encoders for topic segmentation. Specifically, we explore the use of NSEs as feature extractors, which can extract high-level representations leading to better performance on behalf of a topic segmentation system. We investigate what kind of such representations (if any) is most suitable for the task and draw comparisons with existing benchmarks measuring performance on text similarity and related tasks. Finally, we investigate the role of inter-segmental lexical cohesion under different text representations in providing a good segmentation, in three different domains in which topic segmentation has traditionally been investigated: text articles, news broadcast transcripts and business meeting transcripts. The research questions informing this work are:

**RQ1:** Do NSEs lead to better performance than previously used features in topic segmentation?

**RQ2:** Which NSEs work best for this task? Is their ranking consistent with what existing literature has noted about their general performance and how it varies across domains?

**RQ3:** Is the success (or failure) of NSEs always a consequence of better lexical cohesion modelling by the encoders?

While answering the first two questions requires a survey of different encoders, the third one involves the creation of new techniques to quantify lexical cohesion under different encoders and datasets. We do that by introducing a new metric, Average Relative Proximity (ARP). We show that NSEs often do improve over previously used features, but in certain domains improvements are not always significant; and NSEs that are successful in a variety of text similarity tasks do not necessarily improve performance in topic

segmentation. We investigate and explain this with the fact that the factors behind topic segmentation success are not necessarily correlated with a better capacity of identifying cohesive segments and, in certain domains, topic segmentation can rely much more on local cues, making it more similar to a text classification task.

# RELATED WORK

## Models for topic segmentation

Traditionally, topic segmentation involves the segmentation of text such as books or articles (*Beeferman, Berger & Lafferty, 1999*; *Koshorek et al., 2018*), and transcripts of business meetings (*Purver et al., 2006*) or TV news shows (*Misra et al., 2010*; *Sehikh, Fohr & Illina, 2018*).

One of the earliest techniques for topic segmentation, TextTiling, used two adjacent sliding windows over sentences and compared the two by means of cosine similarity between the relative bag-of-words vector representations (*Hearst, 1994*). The same algorithm was then successfully used with different, more informative sentence representations, such as Term-Frequency Inverse-Document-Frequency (TF-IDF) rescoring of bag-of-words (*Galley et al., 2003*) and features derived from generative topic models like Latent Dirichlet Allocation (LDA, *Riedl & Biemann, 2012*). More recently, these topic features have been replaced with sentence representations extracted from large language models (LLMs), again apparently showing improvements (*Ghinassi, 2021*; *Harrando & Troncy, 2021*; *Solbiati et al., 2021*).

Another early approach used the distance between sentence representations in a dynamic programming framework. Similarly to TextTiling, initial work used bag-of-words sentence representations (*Utiyama & Isahara, 2001*), with later literature substituting them with representations from topic models (*Misra et al., 2011*; *Sun et al., 2008*; *Choi, Wiemer-hastings & Moore, 2001*). Improvements over TextTiling have been variously proposed, either by using different scoring of similarities (*Choi, 2000*) or by employing dynamic programming (*Kehagias, Pavlina & Petridis, 2003*; *Eisenstein & Barzilay, 2008*). More recently, attempts to reframe unsupervised topic segmentation as a next-sentence prediction task with pre-trained LLMs have showed major improvements (*Xing & Carenini, 2021*).

With the increasing availability of bigger datasets annotated for topic segmentation, supervised systems have also started to be proposed. *Koshorek et al. (2018)* trained a hierarchical, bidirectional long-short term memory (BiLSTM) neural network to segment paragraphs in a large Wikipedia *corpus*, showing good improvements over non-neural and unsupervised methods. Since then, most literature has focused on using hierarchical recurrent neural networks (*Tsunoo, Bell & Renals, 2017*; *Lukasik et al., 2020a*; *Sehikh, Fohr & Illina, 2018*) or, more recently, hierarchical transformers (*Lukasik et al., 2020b*; *Glavaš & Somasundaran, 2020*). In recent works, the transformer-based BERT (*Devlin et al., 2019*) used as a sentence encoder has been included either to instill additional general knowledge to end-to-end systems (*Xing et al., 2020*) or to extract standalone features (*Lo et al., 2021*).

## Neural sentence encoders

There is no universally agreed term to refer to models that can encode text spanning multiple words, but here we refer to all such models under the umbrella term of Neural Sentence Encoders (NSEs), which has been often used in the context of pre-trained models for tasks such as semantic retrieval, question answering and, generally, tasks for which a single output is produced for one or multiple sentences (*Reimers & Gurevych, 2020a*; *Conneau et al., 2017*; *Cer et al., 2018*). This does not necessarily mean that all of the presented methodologies can be applied just on sentences and in fact these models are often used to encode text spanning multiple sentences or shorter units. What is common for all the discussed NSEs, then, is their ability of encoding in a single dense representation text spanning multiple words.

For many bigger than word-level tasks, in fact, neural models require suitable text representations. Early approaches used simple averaging of word vectors in a sentence or paragraph to give a representation of their semantics, and this can perform quite well in many tasks (*Renter, Borisov & Rijke, 2016*). More recently, a number of specific neural models have been proposed, and such an interest in models has arisen that specific benchmarks have been developed to evaluate their performance on a variety of tasks (*Conneau & Kiela, 2019*; *Wang et al., 2018*).

Among the most successful approaches, the Universal Sentence Encoder (*Cer et al., 2018*) provides sentence embeddings from a deep averaging network (*Iyyer et al., 2015*) pre-trained on Wikipedia and the Stanford Natural Language Inference (SNLI) *corpus* (*Bowman et al., 2015*). More recent efforts directly used BERT (*Devlin et al., 2019*) and similar transformer-based LLMs as sentence encoders (*Li et al., 2020*), or fine-tuned them to output better sentence embeddings by using the distance between semantically related pairs of sentences as an additional training objective (*Reimers & Gurevych, 2020a*).

Especially, these latest transformer-based encoders have shown state-of-the-art results in tasks ranging from semantic textual similarity to natural language inference. Very recently an interest in using such encoders for topic segmentation has also emerged, as opposed to obtain sentence representations in an end-to-end fashion (*Xing & Carenini, 2021*; *Lukasik et al., 2020a*).

Recent works have used NSEs in the context of the unsupervised TextTiling algorithm (*Ghinassi, 2021*; *Harrando & Troncy, 2021*; *Solbiati et al., 2021*) or more advanced unsupervised methods (*Xia et al., 2022*), while a few also investigated supervised models built on top of BERT or other LLMs used as sentence encoders (*Lukasik et al., 2020a*; *Lo et al., 2021*). The most recent literature has followed this latter direction experimenting with efficient fine-tuning of NSEs for the task (*Glavaš, Ganesh & Somasundaran, 2021*) and using pairs of sentence representations directly for topic boundary classification (*Lee et al., 2023*).

As most recent works used just one or two sentence encoders in their experiments, a gap exists in the literature: no direct comparison exists between the influence of different sentence encoding techniques on topic segmentation performance under a common framework.

## Text similarity and topic segmentation

The majority of machine learning techniques for topic segmentation are based on the assumption that the similarity between units of text, such as sentences, will be stronger when they are part of the same topic and will weaken as the topics change (*Reynar, 1999*). This idea aligns with theories in discourse structure that see lexical cohesion and collocation as indicators of local coherence (*Halliday & Hasan, 1976*), and it also connects topic segmentation with other tasks, such as semantic text similarity.

Studies have demonstrated the relationship between text similarity and topic segmentation by evaluating the effectiveness of text similarity metrics as tools for evaluating topic segmentation systems (*Mohri, Moreno & Weinstein, 2009*). This strong correlation is what has justified the use of pre-trained NSEs and neural word embeddings in current literature (*Lo et al., 2021*; *Alemi & Ginsparg, 2015*; *Solbiati et al., 2021*; *Glavas, Nanni & Ponzetto, 2016*); by virtue of their pre-training, dense neural representation at the word and, especially, at the sentence or paragraph level are in fact optimised to represent texts pertaining to the same semantic field closer in the embedding space, and this quality can be variously employed by systems modelling local coherence such as topic segmenters (*Glavas, Nanni & Ponzetto, 2016*). Additionally, some recent works have attempted to directly incorporate the notion of text similarity and lexical cohesion into their models by using multitask learning (*Lo et al., 2021*; *Xing et al., 2020*; *Glavaš & Somasundaran, 2020*).

On the other hand, previous research using hand-engineered features, such as cue words, has also shown that these more specific features are useful in identifying topic shifts in domains such as business meetings (*Joty, Carenini & Ng, 2013*). It is unclear from existing literature which of these two factors, or what combination of the two, is more relevant when applying recent NSE-based techniques to different domains.

In our experiments, then, we also aim to determine the lexical cohesion of same-topic segments under different NSEs and compare how this impacts topic segmentation performance in different domains.

# METHODOLOGY

## Basic units for topic segmentation and general pipeline definition

Previous work has variously used words, sentences, utterances or paragraphs as the input for topic segmentation systems. Recent work has mostly settled on the use of sentences as the basic units for monologue text (*Xing & Carenini, 2021*; *Lo et al., 2021*; *Lukasik et al., 2020a*; *Glavaš & Somasundaran, 2020*; *Koshorek et al., 2018*), and utterances for multi-party conversation (*Xing et al., 2020*; *Solbiati et al., 2021*; *Xia et al., 2022*). Sentences and utterances, in fact, can be conveniently extracted *via* rule-based tokenizers or speaker diarization systems and they have the advantage over word-level segmentation of creating a less sparse setting for topic segmentation and alleviating the natural class imbalance involved in this problem (where the no boundary class is far more represented than the topic boundary one). At the same time, using sentences or utterances is advantageous when using text representations such as those of interest here, as such units of text are closer to the original pre-training settings of such models, which often are not capable of encoding extremely long chunks of text. In this work then we adopt this setting and we

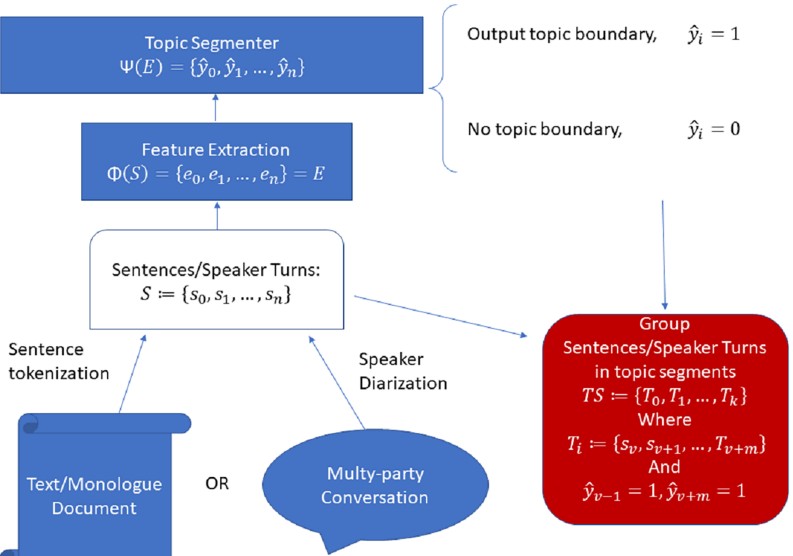

**Figure 1 General pipeline for sentence/speaker turn-level automatic topic segmentation.**

predict whether a topic boundary occur at the level of each individual sentence (in the case of monologue datasets) or speaker turn (for dialogue)—see details in the Dataset section.

This approach comes with the obvious limitation of potentially missing cases in which there might be more than one topic shift inside a single sentence or speaker turn. In all domains we consider, however, and in the majority of work on topic segmentation the granularity of the segmentation is much coarser, as we aim to individuate thematic units that typically span multiple sentences. The annotations for all the datasets we used, therefore, never included multiple topic boundaries in a single sentence, justifying the use of sentences as basic units.

In general, Fig. 1 shows a typical pre-processing and modelling pipeline for a modern topic segmentation system, which is the pipeline we adopted in all our experiments.

In such a pipeline, the input (a monologue text or a multi-party conversation transcript) is first tokenized into sentences or speaker turns $S := \{s_0, s_1, \ldots, s_n\}$, which serve as input to a feature extractor, $\Phi$. This feature extractor can be as simple as a one-hot encoder (as in the original TextTiling algorithm), a NSE or any other suitable function which can vectorize the input text such that $\Phi(S) = \mathbf{E} := \{\mathbf{e}_0, \mathbf{e}_1, \ldots, \mathbf{e}_n\}$, where $\mathbf{e}_i$ is the vectorized form or embedding of the input text $s_i$. Such embeddings serve as input for a machine learning model, $\Psi$. This model outputs a series of binary decision labels $\Psi(\mathbf{E}) := \{\hat{y}_0, \hat{y}_1, \ldots, \hat{y}_n\}$, where a topic boundary occurs at $s_i$ when $\hat{y}_i = 1$. A series of $k$ topic segments $TS := \{T_0, T_1, \ldots, T_k\}$ are then obtained such that each $T_i = \{s_v, s_1, \ldots, s_{v+m}\}$ where $\hat{y}_{v-1} = 1$ and $\hat{y}_{v+m} = 1$, as the sentences enclosed between two positive topic boundary labels constitute a topic segment.

Specifically, we have used the popular punkt sentence tokenizer (*Bird & Klein, 2009*) to extract sentences for our monologue datasets (Wikisection and BBC), while we have used

the provided speaker turns as basic units when using the multi-party conversational dataset (QMSUM).

## Neural sentence encoders

As mentioned, previous works experimented with some NSEs in the context of topic segmentation. These early attempts did not focus on explicitly comparing different encoders; to do that, in this work we compare 13 different NSEs of which four pre-date large transformer-based language models, six are derived from the popular transformer encoder BERT (*Devlin et al., 2019*) and its optimized version RoBERTa (*Liu et al., 2019*), while the last 3 are versions of RoBERTa further fine-tuned on semantic similarity, natural language inference and topic segmentation tasks respectively.

The last encoder, specifically, consists of the RoBERTa pre-trained model which we further fine-tuned for topic segmentation and constitutes a novel contribution in terms of NSE[1], while for all the other models we used the original implementations and pre-trained weights[2]. The encoders we use are described in more detail below:

**Universal Sentence Encoder V4 (DAN):** This architecture from the universal sentence encoders' family consists of a deep averaging network (DAN) that simply averages word embeddings as an input for sentence-level pre-training. This network has been pre-trained on the tasks and with the methodologies described by *Cer et al. (2018)* and briefly mentioned above. The model proved to be very competitive in a variety of benchmarks for semantic text similarity, outperforming BERT and static word vector averaging.

**Universal Sentence Encoder V5 (USE):** This architecture from the universal sentence encoders' family consists of a transformer encoder pre-trained in the same way as the more compact DAN architecture (*Cer et al., 2018*). We abbreviate this architecture as USE to distinguish it from the DAN variant.

**Infersent (Inf):** The infersent architecture was proposed by *Conneau et al. (2017)* as one of the first attempts to create general-use sentence encoders. It consists of a BiLSTM network with max pooling over GloVe word embeddings and it has been pre-trained on a natural language inference task, which proved to be beneficial for training sentence encoders (see also SimCSE below).

**Average GloVe embeddings (GloVe):** This is a very common baseline for sentence encoders that simply averages word embeddings inside a sentence, in this case using GloVe word embeddings (*Pennington, Socher & Manning, 2014*). This approach has been shown to be a good baseline on different benchmarks, managing to beat BERT CLS token (see later) on many tasks, while performing significantly worse than universal sentence encoders and fine-tuned transformers like Para-xlm (below) (*Reimers & Gurevych, 2020a*).

**BERT CLS token ($BERT_{cls}$):** This encoder uses the native sentence encoding capability of BERT: the CLS token that is included at the beginning of each sentence during tokenization. Using this token improved results in many sentence-level tasks, as reported by the original article (*Devlin et al., 2019*). However, the adequacy of this token alone to encode the semantic meaning of a sentence has been questioned by *Reimers & Gurevych (2020a)*, who provided empirical evidence to show that fine-tuning BERT was desirable for semantic text similarity tasks, and that otherwise BERT performance on multiple

[1] The models, each pre-trained on the respective target dataset, are publicly available at: https://huggingface.co/ighina/roberta_topseg_wiki (pre-trained on Wikisection), https://huggingface.co/ighina/RoBERTa_TopSeg_BBC (pre-trained on BBC3000) and https://huggingface.co/ighina/RoBERTa_TopSeg_QMSUM (pre-trained on QMSUM).

[2] We used the sentence_transformers library for extracting para-xlm and GloVe-based sentence embeddings (https://www.sbert.net/index.html), for BERT and RoBERTa, the torch implementation from huggingface was used (https://huggingface.co/docs/transformers/model_doc/bert), for DAN and USE the implementation from tensorflow_hub (https://tfhub.dev/google/universal-sentence-encoder/4), while for infersent and SimCSE we used their official implementation respectively (infersent: https://github.com/facebookresearch/InferSent; SimCSE: https://github.com/princeton-nlp/SimCSE).

benchmarks is even worse than the simple averaging of GloVe embeddings: we would therefore expect this encoder to perform the worst.

**BERT first-last layer average** ($BERT_{first-last}$): In this configuration, we followed *Huang et al. (2021)* and used the average of the first and last layer of BERT for each input sentence as sentence representations. *Reimers & Gurevych (2020a)* already reported that in the context of sentence encoding the average of all token embeddings output by BERT yielded better results than using the CLS token, but *Huang et al. (2021)* further showed that averaging the first and last layer (rather than just the last) yielded better, more stable results. In this context we want to see if BERT's CLS token is consistently worse also in topic segmentation.

**BERT second to last average** ($BERT_{second2last}$): To match the setting of *Xing & Carenini (2021)* we include this pooled representation from BERT that averages the second to last layer and it is suggested as a better sentence representation by the authors of bert-as-service software (https://bert-as-service.readthedocs.io/en/latest/).

**RoBERTa CLS token (RoBERTa):** RoBERTa has been proposed as a robustly optimised version of BERT, capable of outperforming the original BERT in a variety of tasks (*Liu et al., 2019*). Here we compare the CLS token obtained from RoBERTa with the one obtained *via* standard BERT.

**RoBERTa first last average** ($RoBERTa_{first-last}$): As with BERT, we experiment with *Huang et al. (2021)*'s pooling strategy of averaging first and last layers, to compare with the standard CLS token.

**RoBERTa second to last average** ($RoBERTa_{second2last}$): As with BERT, we include the average of the second to last layer, highlighted as a better representation by the authors of BERT-as-a-service.

**Paraphrase-xlm-r-multilingual-v1 (Para-xlm):** This model derives from RoBERTa, but here the sentence representation is obtained by averaging all output tokens from the last layer of the RoBERTa model. The model is further fine-tuned on a dataset of paraphrases to make vectors of paraphrases closer to each other in the embedding space, as described by *Reimers & Gurevych (2020a)*. This encoder also has the advantage of being able to produce embeddings for more than 50 languages, thanks to the additional knowledge distillation described by *Reimers & Gurevych (2020b)*. Encoders using this pre-training approach have been shown to improve over previous encoders such as universal sentence encoder on standard benchmarks for semantic text similarity evaluation (*Reimers & Gurevych, 2020a*). This encoder is the best among the ones we experimented with according to these standard evaluations.

**SimCSE RoBERTa (SimCSE):** This encoder is a version of RoBERTa architecture fine-tuned with a contrastive unsupervised objective described by *Gao, Yao & Chen (2021)*. The sentence information are summarised by the CLS token of the original RoBERTa, which is fine-tuned with the objective of making entailing sentences closer in space and contradicting ones further apart, *via* the contrastive objective described by *Chen et al. (2020)*. We used the pre-trained model from the official implementation: https://github.com/princeton-nlp/SimCSE. By using SimCSE and Para-xlm we aim to test whether those

**Table 1 Sentence encoders used and relative characteristics.**

| Encoder | GloVe | InferSent | USE/DAN | BERT | ROBERTA | Para-xlm | SimCSE | TopSeg |
|---|---|---|---|---|---|---|---|---|
| Embedding size | 300 | 4,096 | 512 | 768 | 768 | 768 | 768 | 768 |
| Train domain | General | General | General | General | General | PR | NLI | In-domain |

pre-training strategies that improve performance of sentence encoders for a variety of other tasks (*Reimers & Gurevych, 2020b*; *Gao, Yao & Chen, 2021*) can also lead to better performance in topic segmentation.

**RoBERTa topic segmentation (TopSeg):** We further include a NSE specifically pre-trained for topic segmentation in the target domain. To do so, we employed the training set of each dataset we used to fine-tune RoBERTa with the objective of making sentences (or speaker turns in the case of QMSUM) from the same topic segment closer in the embedding space. This encoder draws from the evidence collected by *Xing & Carenini (2021)* and *Glavaš & Somasundaran (2020)*, who added an additional loss to their topic segmentation system to directly model the fact that embeddings from the same segment should be closer in space as opposed to embeddings from different segments. *Glavaš & Somasundaran (2020)* achieved this by randomly shuffling the input sentences and having a disciminator recognising the shuffled input together with the topic boundaries, while *Xing et al. (2020)* added a discriminator predicting whether two consecutive segments were supposed to be in the same segment or not. In both cases, the authors showed improved performance resulting from the additional loss, suggesting that this was a consequence of the model learning some notion of local cohesion (in both cases, they referred to it as coherence). In our case, we disentangle the task of having the model predict whether two sentences came from the same segment from the downstream task of topic segmentation. We did so by fine-tuning RoBERTa first on the objective of representing sentences from the same segment closer in space and sentences at the boundary of two different segments further in the embedding space. We then froze the resulting fine-tuned RoBERTa model and, as with the other NSEs, we used the features extracted from the pre-trained encoder to train the topic segmentation system. In fine-tuning, we minimised the following loss:

$$\mathcal{L} = \left\| label_{(i;i+1)} - \frac{\mathbf{e}_i \cdot \mathbf{e}_{i+1}}{||\mathbf{e}_i||_2 \cdot ||\mathbf{e}_{i+1}||_2} \right\|_2 \tag{1}$$

where $\mathbf{e}_i$ and $\mathbf{e}_{i+1}$ are two adjacent embeddings corresponding to sentences $i$ and $i+1$ respectively and $label_{(i;i+1)} = 0.5$ if they both belong to the same segment, otherwise $label_{(i;i+1)} = -1$. The values of 0.5 and $-1$ were used in order to give higher weight to the cases in which embeddings belong to different segments, as these cases represent the minority class.

Table 1 summarize the information of the different encoders used in our experiments, including information about their training domains and their embedding sizes.

### NSE-based topic segmenters

Below we present the topic segmentation models using NSEs, alongside with a description of how they work and (if applicable) how they were trained.

#### Unsupervised topic segmentation system

**DeepTiling** (*Ghinassi, 2021*): this system implements the classic TextTiling algorithm, but using neural sentence embeddings instead of the bag-of-words features used in the original implementation.

Having a NSE $\Phi$ and an input document consisting of a sequence of sentences or speaker turns $S := \{s_0, s_1, ..., s_n\}$, we first compute each embedding $\mathbf{e}_i = \Phi(s_i)$, then having a fixed parameter $w$, we compute a left context for sentence $i$ as:

$$\mathbf{bl}_i = \frac{\sum_{n=i-w+1}^{i} \mathbf{e}_n}{w} \tag{2}$$

(that is, the average of $w$ embeddings on the left of the given sentence); and, similarly, a right context:

$$\mathbf{br}_i = \frac{\sum_{n=i+1}^{i+w} \mathbf{e}_n}{w} \tag{3}$$

Given these two contexts, we compute a score:

$$score_i = \frac{\mathbf{bl}_i \cdot \mathbf{br}_i}{||\mathbf{bl}_i|| \cdot ||\mathbf{br}_i||} \tag{4}$$

Having all such scores we compute so-called depth scores for each one as:

$$ds_i = \frac{score_l + score_r + 2score_i}{2} \tag{5}$$

where $score_l$ is defined as the first peak score on the left of the current score $score_i$ and it is found as the first score such that

$$score_{l_1} < score_l > score_{l+1} \tag{6}$$

with $l < i$, while $score_r$ is the first peak score on the right of $score_i$ such that

$$score_{r_1} < score_r > score_{r+1} \tag{7}$$

with $r > i$.

Finally, a topic boundary is output for each $ds_i > threshold$ where $threshold$ is a custom parameter, as $w$ both of which we tuned on training set. Figure 2 shows the model visually.

#### Supervised topic segmentation system

**NSE-BiLSTM:** As a simple supervised system, we adopted the architecture of *Koshorek et al. (2018)* and substituted the word-level BiLSTM network with NSE.

Having one of the encoders $\Phi$ described above and defining *BiLSTM* as a stack of $n$ BiLSTM layers yielding vectors of dimension $h$, *Softmax* as the softmax function and

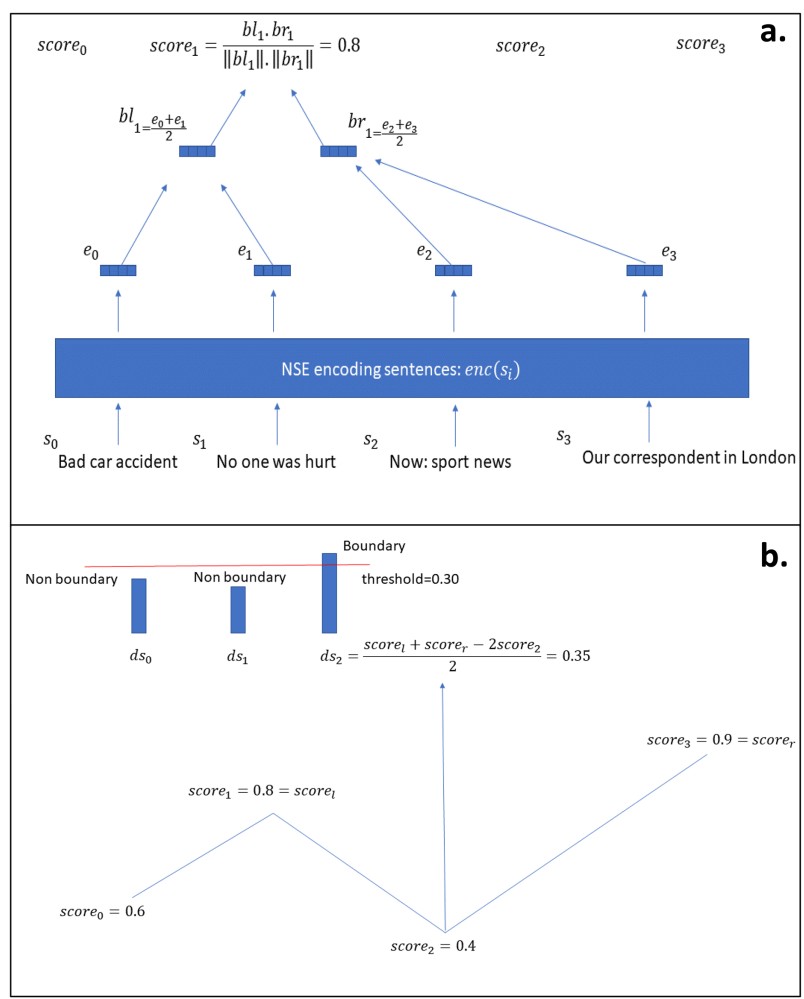

**Figure 2 DeepTiling model.** (A) Shows the process of finding the scores for sentence 1, while (B) shows the process of computing the depth scores from the scores from figure (A) and then using them to output a decision about topic boundaries in case the score is bigger than the given threshold.

$\mathbf{W} \in R^{h \times 1}$ being the weights of the final classification layer we compute the posterior probabilities of each input

$$\hat{\mathbf{y}} = Softmax(\mathbf{W}^T BiLSTM(\mathbf{E})) \tag{8}$$

where $\mathbf{E} := \{\mathbf{e}_0, \mathbf{e}_1, ..., \mathbf{e}_n\}$ is the sequence of embeddings, each corresponding to a sentence, as extracted by the current encoder $\Phi$ and the probabilities $\hat{\mathbf{y}} := \{\hat{y}_0, \hat{y}_1, ..., \hat{y}_n\}$ represent the probabilities of being the end of a topic segment that the model attributed to sentences 0 to $n$ respectively.

Each sentence encoder $\Phi$ is not fine-tuned, acting just as a static feature extractor.

An output boundary is outputted when $\hat{y} \geq threshold$: where $0.05 \leq threshold \leq 0.95$ and $threshold$ is optimised by grid-search on validation set during training.

Figure 3 shows the proposed model visually.

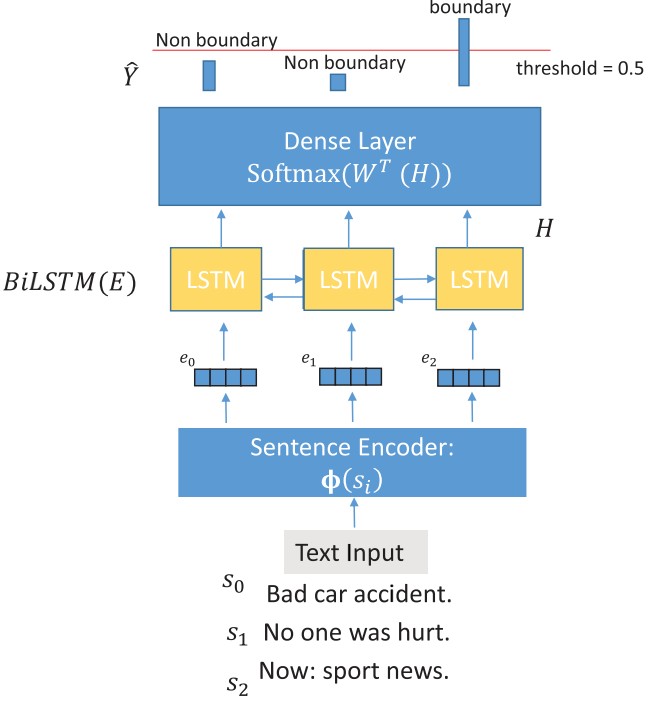

**Figure 3** **NSE-BiLSTM model architecture.**

## Non-NSE-based topic segmenters

For completeness we also evaluate how our NSE-based models perform compared to other models having the same structure but non-neural or word-level input features. Specifically we include the following Non-NSE baselines:

### Unsupervised topic segmentation systems

**TopicTiling** (*Riedl & Biemann, 2012*): An alternative version of TextTiling using the topic ID assigned to each word with highest probability by an LDA model to build the sentence vectors.

   **TextTiling** (*Hearst, 1994*): The original TextTiling algorithm. The main difference with DeepTiling and TopicTiling is that the input features are sparse vectors with absolute word counts in each sentence.

### Supervised topic segmentation system

**TextSeg** (*Koshorek et al., 2018*): This model is the same as NSE-BiLSTM but for a word-level BiLSTM network modelling Word2Vec word embeddings (*Mikolov et al., 2013*) to create sentence representations. This lower-level network passes each word embedding in a sentence to two recurrent layers with 256 hidden units and then applies a max pooling operation over the encoded input to obtain a sentence representation. The sentence-level BiLSTM, then, is identical to NSE-BiLSTM.

   Finally, we include one random and one rule-based baseline:

   **Binomial:** Using the same setting as *Koshorek et al. (2018)* we included a random baseline that outputs a sentence-level decision over boundaries based on a binomial

distribution having $\frac{1}{k}$ probability of success (*i.e.*, a boundary is output), where $k$ is the average length of segments in the dataset.

**Cue Word:** This baseline is a simple rule-based method. We automatically detect the ten most frequent words from the final sentence (excluding function and stop words) of each segment in the training set and we then derive a rule to create a new segment where at least one of these words is detected. This method, as we will see, can be surprisingly effective in scenarios in which segments tend to end with a formulaic expression, such as those appearing in certain news shows.

## Datasets

We used three datasets covering a range of domains: face-to-face meetings (QMSUM), written articles (Wikisection) and TV news broadcasts (BBC3000). The datasets have been chosen to cover the type of domains in which topic segmentation has been traditionally investigated: the conversational domain, where usually datasets extracted from business meetings have been used (*Janin et al., 2003*; *Solbiati et al., 2021*; *Xia et al., 2022*); the books or articles domain, which is the most usual domain of experimentation in the field given the convenience of automatically obtaining datasets by scraping articles from Wikipedia (*Koshorek et al., 2018*; *Arnold et al., 2019*); and the media domain, where most work has focused on segmentation of structured radio or TV programmes such as news shows (*Ghinassi et al., 2023*; *Wang et al., 2012*; *Sehikh, Fohr & Illina, 2018*). The three datasets we used and that we describe in more detail below, then, can be seen as representative of these three domains, as all but the last are very popular datasets in the respective literature, while the last dataset represents the media domain, where datasets are usually private due to copyright issues.

**QMSUM** (*Zhong et al., 2021*): This dataset was introduced in the context of meeting summarization and aggregates three smaller datasets from multiple domains: ICSI (*Janin et al., 2003*), a dataset of real-world academic meetings, AMI (*Carletta et al., 2006*), a *corpus* of simulated business meetings, and a third dataset of parliamentary committees meetings from Wales and Canada's parliaments, that was newly annotated by the authors. As such, the dataset covers a range of domains, but all in a conversational setting, which proved to be more challenging for topic segmentation (*Solbiati et al., 2021*). The dataset includes 232 meetings and provided a standard training, validation and test set. Each meeting is annotated with summaries and, crucially, with the topic currently being discussed. From the topical annotation, topic boundaries can then be derived to train a topic segmentation system, as shown in the original article (*Zhong et al., 2021*). Speaker turns are provided by the author and we use the utterances thus defined instead of sentences when using this dataset.

**Wikisection** (*Arnold et al., 2019*): Wikisection was recently proposed as a dataset for topic segmentation and classification in both German and English. The dataset was obtained by scraping Wikipedia articles from two different domains: medical and geographical (specifically, articles about cities). The ground truth topic boundaries are obtained by exploiting the headers present in the original HTML files. We follow *Xing & Carenini (2021)* and we consider only topic boundaries occurring between headers of level

**Table 2 Statistics from the datasets used.**

| Statistics | QMSUM | Wikisection | BBC3000 |
|---|---|---|---|
| Num. Segs | 1,286 | 90,048 | 32,074 |
| Av. Seg length | 96.93 ± 6.11 | 14.00 ± 0.12 | 22.00 ± 0.19 |
| Segs per Doc | 5.54 ± 0.26 | 3.93 ± 0.03 | 10.77 ± 0.11 |

1 and 2 (therefore excluding any boundary between sub-topics). We used just the English partition of the dataset and exclude any article not having topic boundaries when considering just the headers described before.

**BBC3000:** This is a private dataset consisting of 3,077 transcripts from news shows aired on the BBC news channel between 2014 and 2020. The training set consists of 2,707 such documents, while the test set includes the last 270 shows. In addition to the already provided train-test split, we used 270 shows from the train set to create a validation set, by extracting one show every 10. The transcripts were created manually by expert annotators and, therefore, they are reliable in terms of both content and punctuation.

Table 2 shows some statistics from the three datasets used. Notice how QMSUM has extremely long segments compared to the other two datasets, while Wikisection is by far the biggest dataset.

## Evaluation

Most works in the field, use the $P_k$ metric (*Beeferman, Berger & Lafferty, 1999*) to evaluate topic segmentation models. $P_k$ gives a measure which does take into account how close the segmentation is to the ground truth on average, but has been reported to be untrustworthy in a number of contexts (*Georgescul, Clark & Armstrong, 2006*) and, as such, better options such as WindowDifference (*Pevzner & Hearst, 2002*) has been developed during the years. Existing literature still mostly use the outdated $P_k$ metric. To have a more trustworthy evaluation, we used the Boundary Similarity metric developed by *Fournier (2013)*. Boundary Similarity employs the well-known minimum edit distance algorithm to obtain the minimum operations required to get from a hypothetical segmentation to the reference one. Each operation has a specific weight, where transposition of nearby boundaries are not penalised as bad as addition, deletion or direct substitution of an incorrect boundary. We keep the default value of 2 for the parameter controlling how close an incorrect boundary has to be in order to be considered a near miss.

## EXPERIMENTAL SETUP

Each of the three datasets have been split into a training, validation and test set and each model we present is trained and tested independently on each dataset using the corresponding partitions; when we report results for NSE-BiLSTM for QMSUM, for example, such results were obtained from the test set of that dataset with a NSE-BiLSTM model trained on QMSUM's training set and optimised on QMSUM's validation set. The same applies to all baselines (more details for the individual models are found below).

For every NSE-BiLSTM and TextSeg experiment we used 256 hidden units and two recurrent layers, to match the original configuration of *Koshorek et al. (2018)*. Learning rate was set to 0.001 and the Adam optimization algorithm (*Kingma & Ba, 2015*) was used with a scheduler reducing the learning rate by a factor of 0.1 in case of no improvement on validation set loss over 10 epochs. A batch size of 8 was used for every experiment and we stopped training after 20 consecutive epochs without improvements on validation set's boundary similarity.

For the unsupervised systems, we have found the optimal values for threshold and window size on the training set of each dataset and encoder *via* grid-search over the following sets of parameters: $window \in \{1, 5, 15, 25, 75\}$ and $threshold \in \{0.5, 1.5, 2.0, 2.5\}$.

For TopicTiling, we first fitted a topic model with 50 hidden topics on each respective training set and then used that to obtain posterior probabilities for the words in each dataset. Before fitting TextTiling and TopicTiling, stop words were removed and lemmatization was applied.

Finally, we also tested the hypothesis of the results of the best performing of our models for every dataset being significantly better than the best performing baseline from the relative dataset, by using a two-sample hypothesis testing for mean difference *via* bootstrapping the difference in average test results as explained in *Davison & Hinkley (1997)*, Chapter 4. We report statistical significance with a *p*-value smaller than 0.05 by marking the best result from our models with an asterisk (*) (or not marking it, in case of non-significance).

## BASELINE COMPARISON

Tables 3 and 4 report the results we achieved with our encoders for each dataset and NSE-based model, as well as the baseline results. Table 3 reports results obtained with the supervised systems, while Table 4 shows the results obtained with the unsupervised and the random models.

A general look at the table confirms that the answer to our first question is positive: NSEs seem to outperform all of the baselines, as at least one of the neural sentence encoders proposed always perform better than the baselines based on word-level or non-neural features.

If we compare DeepTiling with the other two unsupervised baselines, TextTiling and TopicTiling, we see that in fact all NSEs perform better than the baselines, except for the very good performance of TextTiling in BBC3000 which places it in 6th position.

In the case of TextSeg, the baseline model is competitive with NSE-BiLSTM in Wikisection and BBC3000, where it beats most non-transformer NSEs. In Wikisection the supervised models using transformer-based encoders lead to improvements over this baseline and the best configuration using TopSeg encoder significantly outperforms TextSeg.

In the case of BBC3000 dataset, instead, using TextSeg leads to mostly better results than NSE-based models and, even though, the best NSE configuration is marginally better than the TextSeg baseline, such an improvement is not significant. This evidence is discussed in

**Table 3 Results of supervised topic segmentation models on QMSUM, Wikisection (Wiki) and BBC3000 (BBC) datasets in terms of boundary similarity for NSE-BiLSTM and TextSeg baseline.**

| Dataset | QMSUM | Wiki | BBC |
|---|---|---|---|
| **NSE-BiLSTM** | | | |
| GloVe | 5.79 | 56.56 | 81.42 |
| DAN | 10.43 | 57.27 | 84.86 |
| USE | 11.42 | 57.63 | 86.05 |
| Inf | 13.88 | 56.04 | 87.28 |
| $BERT_{cls}$ | 11.83 | 60.55 | 88.43 |
| $BERT_{first-last}$ | 14.44 | 59.17 | 87.60 |
| $BERT_{second2last}$ | 12.84 | 59.92 | 88.30 |
| $RoBERTa_{cls}$ | 7.41 | 58.73 | 88.13 |
| $RoBERTa_{first-last}$ | **16.04**[*] | 58.67 | 88.30 |
| $RoBERTa_{second2last}$ | 14.89 | 59.84 | **89.45** |
| Para-xlm | 11.67 | 57.95 | 86.83 |
| SimCSE | 12.07 | 58.40 | 84.91 |
| TopSeg | 14.49 | **61.24**[*] | 87.50 |
| **Supervised baseline** | | | |
| TextSeg | 10.53 | 59.31 | 89.18 |

Notes:
Higher values indicate better results. Bold font indicates best results on the relative dataset.
[*] Indicates results are significantly better than baseline ($p < 0.05$).

more details later, where the specific case of the news shows domain is analysed in more details.

For what concerns the conversational domain, TextSeg performs worse than the majority of the NSE-BiLSTM models but for the ones based on GloVe and, surprisingly, $RoBERTa_{cls}$. Using the average of first and last layers for the same RoBERTa model, instead, leads to the best results, that are significantly better than the baseline. Also if we look at $BERT_{cls}$ and $BERT_{first-last}$ we can confirm that the pooling strategy proposed by *Huang et al. (2021)* leads to significant improvements in the case of conversational data such as QMSUM.

A surprising result is that of fine-tuned transformers for BBC3000. In this case they all underperform with respect to the relative baselines, but for TopSeg being the best NSE in DeepTiling. This evidence will be further explored in the next section.

As a quick note about the random and rule-based baselines, all models seem to generally perform better than these settings but for the striking results of Cue Word baseline on QMSUM and, especially, BBC3000 where it outperforms most unsupervised systems. This evidence is indeed very interesting and the next sections will expand on this.

# SENTENCE ENCODERS COMPARISON

## NSE-BiLSTM comparison with DeepTiling

When we compare the various sentence encoders and our two models presented in Tables 3 and 4, we can immediately see how NSE-BiLSTM outperforms DeepTiling in every

**Table 4 Results of unsupervised topic segmentation models and random baselines on QMSUM, Wikisection (Wiki) and BBC3000 (BBC) datasets in terms of boundary similarity for our DeepTiling, unsupervised and random baselines methods.**

| Dataset | QMSUM | Wiki | BBC |
|---|---|---|---|
| **DeepTiling** | | | |
| GloVe | 2.96 | 24.02 | 15.69 |
| DAN | 4.65 | 25.34 | 22.04 |
| USE | 4.17 | 25.71 | 23.49 |
| Inf | 3.74 | 22.94 | 13.35 |
| $BERT_{cls}$ | 2.82 | 22.91 | 25.25 |
| $BERT_{first-last}$ | 2.56 | 23.29 | 12.73 |
| $BERT_{second2last}$ | 2.81 | 23.12 | 14.34 |
| $RoBERTa_{cls}$ | 3.51 | 20.53 | 18.03 |
| $RoBERTa_{first-last}$ | 3.26 | 19.92 | 22.49 |
| $RoBERTa_{second2last}$ | 3.26 | 21.73 | 21.65 |
| Para-xlm | 4.93 | 25.75 | 21.48 |
| SimCSE | 4.86 | 27.39 | 21.68 |
| TopSeg | **12.22**[*] | **31.98**[*] | **69.55**[*] |
| **Unsup. Baselines** | | | |
| TopicTiling | 0.63 | 8.39 | 11.87 |
| TextTiling | 2.95 | 16.30 | 21.82 |
| **Other baselines** | | | |
| Binomial | 1.28 | 8.73 | 5.00 |
| Cue word | 3.60 | 11.46 | 26.07 |

**Notes:**
Higher values indicate better results. Bold font indicates best results on the relative dataset.
[*] Indicates results are significantly better than baselines ($p < 0.05$).

setting, proving the superiority of supervised approaches for the task, when training data is available.

## Transformers-based encoders comparison with other architectures

In terms of encoders used, Transformer-based encoders generally outperform the alternatives, in line with existing literature on sentence encoders. GloVe consistently performs the worst when used with NSE-BiLSTM, with the only notable exception of Wikisection, where Infersent is marginally worse even than GloVe.

If we look at the ranking of the encoders for DeepTiling or even at the ranking of Transformer-based encoders for NSE-BiLSTM, however, we realise that results contradict existing literature in many respects.

DeepTiling seems to benefit from GloVe embeddings more than from RoBERTa (Wikisection) or BERT based ones (QMSUM): this seems at odds with the evident superiority of those methods when using a supervised approach, while also contradicting the ranking of these encoding methods in existing benchmarks for sentence encoder evaluation. Different pooling strategies for BERT and RoBERTa do not necessarily improve results when looking at DeepTiling, while NSE-BiLSTM largely confirm that

averaging different layers result in better results than simply using the CLS token; a notable exception to this rule, however, can also be seen in the supervised results of BERT CLS, which is the best among the non fine-tuned encoders.

### Fine-tuned RoBERTa comparison with other encoders

Looking at the fine-tuned RoBERTa models, it can immediately be seen that in general fine-tuning the basic model yields bigger improvements for DeepTiling rather than for NSE-BiLSTM. A striking result is the fact that TopSeg does not perform the best when coupled with NSE-BiLSTM for QMSUM and BBC datasets. As the encoder is specifically fine-tuned on the topic segmentation task we would have expected that setting to always score the best.

This is indeed the case when looking at DeepTiling, that is hardly surprising considering that in fact the cosine distance between adjacent embeddings is at the base both of the pre-training strategy and of the unsupervised algorithm. Using TopSeg with DeepTiling, then, is equivalent to injecting information into the model in a supervised fashion, which obviously gives a great advantage to this setting when compared with the other totally unsupervised ones.

### Dataset comparisons

The discrepancies between literature on neural sentence encoders and the use of the same in topic segmentation are evident in the fact that methods such as SimCSE and Para-XLM do not seem to yield any consistent improvement over the non fine-tuned baselines and this is true both when looking at NSE-BiLSTM and (in a smaller measure) DeepTiling models.

The fact that transformer-based models tend to be better instead conforms to what we could have expected and suggest that maybe a better interpretation of the results can be achieved by thinking at each individual dataset.

The three domains we are looking at, in fact, present notable differences with each other, and the internal segment coherence might have a very different role in a conversational setting such as QMSUM and a text-based benchmark like Wikisection. This is evident when we look more closely at the varying degree of success that a simple rule-based baseline like Cue Word can have in different datasets. The simple baseline performs worse than any other system for a dataset like Wikisection, it is competitive in QMSUM and it actually outperforms every DeepTiling configuration but the TopSeg one for BBC3000.

We hypothesise that part of the reason for this phenomenon lies in the fact that certain domains present more conventional, formulaic ways of terminating a topic than others. In news shows, especially, repetition of the name of the channel or a greeting to the current reporter are both common ways in which a segment might end, while in a meeting various disfluencies such as "um" or fillers such as "right" tend to appear more often towards the end of discussing a topic, as there are fewer things left to say and interlocutors start to consider whether they forgot something (*Purver et al., 2006*).

The same logic highlighting the importance of specific cue words for achieving good segmentation results in certain domains is certainly to be linked to the good results of TextSeg for BBC3000 shown in the previous section. TextSeg is a hierarchical supervised system modelling word-level embeddings. At the same time, it uses a max pooling approach to create sentence representations from words (*Koshorek et al., 2018*) and later research has shown that using an attention-based pooling approach yields considerable improvements (*Xing & Carenini, 2021*). All of this is likely to be related to the fact that hierarchical supervised models benefit from focusing on specific cue words, which, in cases like BBC3000, might make them competitive with models building on semantically richer features like NSE-BiLSTM.

The factors leading to a specific model or neural sentence encoder outperforming another one in topic segmentation, therefore, are not limited to how well that encoder can encode similar texts closer in space, as supervised systems such as NSE-BiLSTM and, especially, TextSeg might actually focus more on individuating specific transition marks, while even a model like DeepTiling might perform well enough thanks to a closing sentence being on average further in the embedding space from other sentences.

In the next section, then, we introduce a novel method to quantify the cohesion of ground truth segments under each dataset and each embedding. By correlating this quantity with our results we aim to isolate the influence of embeddings' coherence on topic segmentation results, so to better answer the point raised above.

# SEGMENTATION COHESION COMPARISON

## ARP scores

Recent work in topic modelling has showed how simple clustering performed on top of sentence embeddings is an effective way of obtaining coherent topics that can in many cases perform better than more complex neural and traditional topic models (*Angelov, 2020*; *Zhang et al., 2022*; *Sia, Dalmia & Mielke, 2020*; *Grootendorst, 2022*). At the same time, we already described how NSEs have been used in the context of topic segmentation with the rationale of having features that can better capture the lexical cohesion indicating a common theme, where a better representation in this case implies encoding topically similar sentences closer in the embedding space (leading to such sentences having a higher text similarity score). Sentence embeddings that are able to correctly represent the underlying semantic field of a common topic, then, would start to deviate from previous ones as the topic change and this is exactly the intuition behind unsupervised approaches such as DeepTiling (*Ghinassi, 2021*).

Therefore, we quantify this phenomenon directly in the embedding space as the total variance of a group of embeddings that are supposed to be in the same segment according to ground truth topic segmentation labels (intra-segmental variance). When compared to the variance of embeddings from different, adjacent segments (inter-segmental variance), this can give us an idea of how well each encoder can embed individual sentences from the same segment to be closer in the embedding space as opposed to sentences from different segments. Secondly, by comparing such a quantity of relative proximity to downstream

topic segmentation results, we can draw conclusions about the importance that encoding same segment closer in space has in different domains.

More formally, we define Average Relative Proximity (ARP) as a metric to evaluate the overall coherence of segments in the embedding space. The method first computes the intra-segmental variance:

$$intravar = ||std(\mathbf{E}_{tra})||^2 \tag{9}$$

where $\mathbf{E}_{tra}$ is a set of embeddings in a same segment. It then computes the inter-segmental variance with the next segment:

$$intervar = ||std(\mathbf{E}_{ter})||^2 \tag{10}$$

where $E_{ter}$ includes embeddings in $\mathbf{E}_{tra}$ and embeddings in the following segment.

In order to overcome the standard deviation bias of getting bigger for smaller samples, we force the intra-variance to be computed on a collection of embeddings having the same length as the ones from which the inter-variance is computed. This is simply done by adding a cutting point $cut = |\mathbf{E}_i|/2$ where $|\mathbf{E}_i|$ is the length of the current segment, so that $\mathbf{E}_{tra} = \mathbf{E}_i$ and $\mathbf{E}_{ter} = \mathbf{E}_i^{cut:|\mathbf{E}|} \oplus \mathbf{E}_{i+1}^{0:cut}$, with $\mathbf{E}_{i+1}$ being the next segment and $\oplus$ representing concatenation.

The two scores are aggregated to obtain a single score representing the relative proximity (RP) of embeddings belonging to the same segment. We compute it as

$$RP = 100 \times \frac{intervar - intravar}{intervar + intravar}. \tag{11}$$

While the constant 100 serves solely the purpose of better visualisation, the normalization factor (*i.e.*, the sum of the two variances) forces the score in the range of $-100$ to 100 and both extremes have a unique interpretation. RP $= -100$ corresponds to the case of maximum distance between embeddings in the same segment and maximum proximity of embeddings in between segments (worst case). RP $= 100$ corresponds to the case of maximum proximity between embeddings from the same segment and maximum distance of embeddings from different segments (best case). RP $= 0$ is when there is no difference between the intra and inter-distance of the analysed embeddings.

The ARP score on a dataset with a specific sentence encoder is then obtained by simply averaging all the relative proximity scores

$$ARP = \frac{1}{N_{seg}} \Sigma_{i=1}^{N_{seg}} RP(i) \tag{12}$$

where $N_{seg}$ is the total number of segments in the dataset.

The ARP method thus defined can highlight the relative cohesion of a sequence of topically coherent segments. The method differs from previous coherence measures used for topic modelling evaluation[3] in that ARP serves the different purpose of highlighting the inner coherence of consecutive document-level topic segments, while other coherence measures usually highlight the internal coherence of individual *corpus*-level topics. As such, this specific coherence score reflects the specific situation of topic *segmentation*,

---

[3] A comprehensive summary of such metrics can be found in *Röder, Both & Hinneburg (2015)*.

where we want to infer the boundaries of existing topic segments, rather than classifying the entire document into (often) overlapping topics underlying the entire *corpus*.

## Dataset comparison

Table 5 shows all the ARP scores for each dataset and encoder we used, using the ground truth to obtain the true segments and compute the various inter and intra-segmental variances. In the table we also include the ARP score obtained with the same setting but using random segments rather than the ground truth ones, while the third column explicitly shows the ranking of each encoder in the relative dataset in terms of ground truth ARP scores.

While the ranking can tell us which encoders seem to be more successful in encoding coherent segments, showing the ARP score for random segments is useful to check that indeed the ARP scores for any given setting are better than random.

In this way we can immediately see that this condition holds true for all the encoders in Wikisection and BBC, but not for the worst performing encoders in QMSUM. This evidence is likely to reflect what we observed in the previous section about this dataset and its conversational nature. In conversation, given the numerous disfluencies and speech acts, we do expect embeddings to be distributed more randomly within the same segment and the low ARP scores are indeed a strong confirmation that this is happening and, as a consequence, the ability of encoders to embed intra-segmental sentences closer in space might be less relevant in this context.

The same can be observed for BBC3000; even though all sentence encoders are able to encode intra-segmental sentences closer in space, resulting in positive and significantly greater than 0 ARP scores, the magnitude of the scores is quite low compared to what we can observe in the case of Wikisection, once more reflecting a lower intra-segmental coherence in that dataset/domain.

## Correlation of ARP scores with topic segmentation results

We can further confirm the above intuition by looking at Fig. 4, where the correlation between our topic segmentation systems and the relative ARP scores is displayed. NSE-BiLSTM and, in a smaller measure, DeepTiling results for Wikisection are all quite strongly correlated with the ARP scores, while this is not so clear for the other two datasets. The correlation between DeepTiling results for BBC3000 and its ARP scores is actually extremely high, while it drops to a very weak correlation for NSE-BiLSTM. Overall, this seems to hint at the fact that a supervised system in this context does not focus much on the coherence of the embeddings' sequence as other information contained in single embeddings might be enough to solve the task satisfactorily (see the high boundary similarity obtained by the rule-based method for this dataset). Appendix A further illustrates this.

QMSUM, instead, consistently shows relatively weak correlation between ARP scores and topic segmentation results, further confirming what we observed before.

**Table 5  ARP scores for every sentence encoder and dataset.**

| Dataset | Encoder | ARP (Ground Truth) | ARP (Random Segment) | ARP ranking |
|---|---|---|---|---|
| Wikisection | GloVe | 4.10 | 0.03 | 6 |
| | DAN | 2.25 | 0.03 | 13 |
| | USE | 2.45 | 0.01 | 12 |
| | Inf | 3.59 | −0.07 | 9 |
| | $BERT_{cls}$ | 5.81 | 0.01 | 2 |
| | $BERT_{first-last}$ | 4.08 | −0.02 | 7 |
| | $BERT_{second2last}$ | 5.77 | −0.01 | 3 |
| | $RoBERTa_{cls}$ | 5.05 | −0.01 | 4 |
| | $RoBERTa_{first-last}$ | 3.33 | 0.07 | 10 |
| | $RoBERTa_{second2last}$ | 4.72 | 0.03 | 5 |
| | Para-xlm | 2.70 | 0.04 | 11 |
| | SimCSE | 3.81 | −0.01 | 8 |
| | TopSeg | 12.15 | −0.26 | 1 |
| BBC3000 | GloVe | 1.12 | 0.06 | 12 |
| | DAN | 1.25 | −0.02 | 10 |
| | USE | 1.39 | −0.02 | 8 |
| | Inf | 1.46 | −0.09 | 6 |
| | $BERT_{cls}$ | 1.56 | −0.03 | 4 |
| | $BERT_{first-last}$ | 1.21 | 0.01 | 11 |
| | $BERT_{second2last}$ | 1.44 | 0.06 | 7 |
| | $RoBERTa_{cls}$ | 1.76 | 0.04 | 3 |
| | $RoBERTa_{first-last}$ | 1.02 | 0.03 | 13 |
| | $RoBERTa_{second2last}$ | 1.50 | −0.07 | 5 |
| | Para-xlm | 1.26 | −0.06 | 9 |
| | SimCSE | 2.00 | −0.03 | 2 |
| | TopSeg | 4.83 | −0.02 | 1 |
| QMSUM | GloVe | −0.10 | 0.11 | 13 |
| | DAN | 0.13 | 0.11 | 10 |
| | USE | 0.10 | 0.15 | 11 |
| | Inf | 1.50 | −0.4 | 1 |
| | $BERT_{cls}$ | −0.09 | 0.10 | 12 |
| | $BERT_{first-last}$ | 0.26 | −0.26 | 8 |
| | $BERT_{second2last}$ | 0.34 | 0.10 | 7 |
| | $RoBERTa_{cls}$ | 0.70 | −0.05 | 3 |
| | $RoBERTa_{first-last}$ | 0.43 | −0.39 | 5 |
| | $RoBERTa_{second2last}$ | 0.53 | −0.31 | 4 |
| | Para-xlm | 0.24 | 0.41 | 9 |
| | SimCSE | 0.31 | 0.02 | 6 |
| | TopSeg | 0.91 | −0.01 | 2 |

**Note:**
The ARP scores (Ground Truth) column shows the ARP scores computed on the ground truth labels, while the ARP Scores (Random Segments) column show the scores computed with the same sentence encoder on randomly segmented input documents. In both cases the higher the ARP score the better a sentence encoder is able to encode same-segment sentences closer in space as opposed to cross-segment ones.

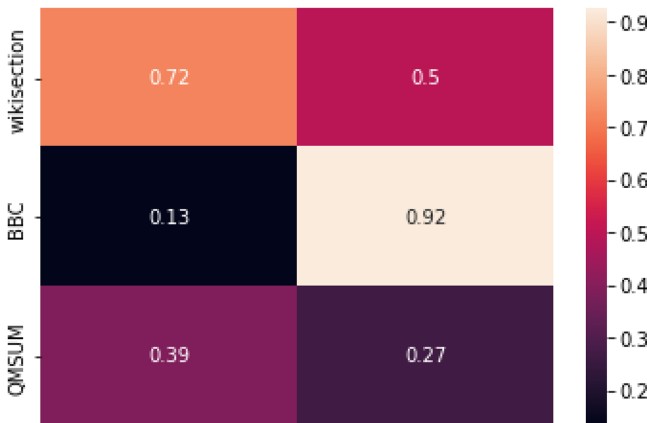

**Figure 4 Correlations between the results from NSE-BiLSTM and DeepTiling and the relative ARP scores computed on ground truth for each sentence encoder.** Pearson correlation coefficient has been used as correlation score.

## Encoder rankings

Finally, we can look at the ranking of the encoders by their relative ARP scores and draw some conclusions about these as well, while keeping in mind the bigger picture just described.

In almost every case, TopSeg scores the highest as it has been directly trained for representing coherent segments in the target domain. We can see that the other two fine-tuned encoders, Para-xlm and SimCSE, have varying success in the same task, as their ARP scores are quite low for BBC3000 and QMSUM. A very surprising result is that of Infersent placing itself first for QMSUM; even though this would be worth further exploration, this can also be partly explained by the fact that scores are anyway always very poor in this dataset. The relatively good performance of Transformer-based encoders, however, seems to be confirmed also under this metric, as in every case the top three encoders mostly belong to that category.

If we combine the individual ARP scores with the more general picture that we drew before when looking at the correlation map with segmentation results, two main points can be made:

1) Existing sentence encoders pre-trained on large corpora for textual similarity do not transfer the same good performance to topic segmentation. This is evident by the non-convincing performance of Para-xlm and SimCSE; while TopSeg trained on the target domain is able to perform much more convincingly, still simple methods like CLS token from BERT or RoBERTa perform competively with respect to more *ad hoc* models. If we look at the ARP scores in conversational domains, then, it becomes evident that sentence encoders fine-tuned for textual similarity fail to take into consideration the longer context and probably to disregard the redundant information that multi-sentence segments might present, leading to a limited ability to encode same segments closer in a coherent embedding space.

2) Even overcoming the limitations described above would not guarantee better segmentation results, as in certain settings the problem of topic segmentation can partly be solved by text classification. Where specific cues are present that signal a potential topic boundary, the importance of encoding the more global coherence of the underlying text becomes less relevant.

Finally, ARP scores help analysing the encoders under a different light, but could also work as feature selector as they successfully place in the top three encoders the ones that end up having the best topic segmentation results 5 out of 6 times. Future work might explore further the potential of ARP scores in pre-selecting the correct sentence encoders without the need for any additional training; this direction is especially worth considering for domains such as written articles like Wikisection, where ARP scores are strongly correlated with downstream segmentation performance.

### Specificity of topic segmentation as a text similarity task

The evidence thus collected has shown that text similarity benchmarks commonly used in much literature advancing the state of the art of NSEs do not reflect the performance of those same NSEs on the task of topic segmentation. The ARP scores we defined in this section further confirm this evidence: even though the score itself is closely related to text similarity, in fact, it is clear that SimCSE and Para-xlm, both of which have shown good improvements on standard text similarity benchmarks (*Reimers & Gurevych, 2020a*; *Gao, Yao & Chen, 2021*), do not perform any better than other encoders.

From a qualitative point of view, we can explain this fact *via* a number of different factors:

1) Regarding performance on QMSUM, standard benchmarks typically do not involve conversational data.
2) Notwithstanding this limitation, the semantic granularity needed to solve standard text similarity tasks is usually finer than that needed for topic segmentation. Topic segmentation, in fact, aims to find parts of a text related to a coherent semantic field or domain: even just the fact of identifying such a field is a non-trivial task that might depend on a certain level of world knowledge (*Ferret & Grau, 2000*) and it usually implies semantic connections that are looser than, for example, identifying a set of paraphrases or whether a sentence entails or contradict another, both of which are common pre-training strategies for standard text similarity tasks.

Focusing on this last point, especially, we can qualitatively analyse the similarity assigned by different encoders to specific pair of sentences.

Table 6 shows a number of sentence pairs having different semantic relationships with each others, as well as the cosine similarity score assigned to each pair by different encoders.

From the table we can see that different NSEs tend in fact to manifest a different behaviour in how closely related they consider pairs of sentences. Para-xlm and SimCSE show a behaviour consistent with traditional semantic text similarity tasks, reflecting their

**Table 6 Qualitative examples of the cosine similarity assigned to pair of sentences by different NSEs.**

| Sentence 1 | Sentence 2 | TopSeg | $BERT_{cls}$ | SimCSE | Para-xlm |
|---|---|---|---|---|---|
| **Paraphrases** | | | | | |
| Katz was born in Sweden in 1947. | Katz was born in 1947 in Sweden. | 0.98 | 0.98 | 0.98 | 0.97 |
| **NLI** | | | | | |
| A statue at a museum that no seems to be looking at. | There is a statue that not many people seem to be interested in. | 0.86 | 0.97 | 0.78 | 0.78 |
| **Topic segmentation** | | | | | |
| The highest chimney in Europe is at a power station in Slovenia. | It stretches more than 1,180 feet into the sky. | 0.46 | 0.97 | 0.27 | 0.10 |
| **Unrelated sentences** | | | | | |
| Cars are made of steel | The sea is blue, the grass is green | 0.20 | 0.92 | 0.01 | 0.14 |

Note:
The first pair are paraphrases. The second pair pertain to natural language inference (NLI), where in this case the first sentence entails the second. The third pair consists of two consecutive sentences in a topic segment, while the last pair includes two unrelated sentences.

pre-training and they assign high scores to paraphrases and entailing sentences, while they assign very low scores to the semantically unrelated pair. Both of these encoders assign very low cosine similarity also to the topically related pair, too, reflecting their relatively poor performance in downstream topic segmentation.

On the other hand, $BERT_{cls}$ give a high score to the topically related pair, but it can be seen that it assigns a quite high score to the unrelated pair as well, even though comparatively lower than to the other categories.

The encoder we pre-trained on topic segmentation itself, TopSeg, gives high scores to Paraphrases, NLI and Topic Segmentation pairs and a low score to the unrelated sentences. In particular, it can be seen that even though it still scores paraphrases and even entailing sentences much higher, the score assigned to the pair of sentences from the same topic segment is considerably higher than the one assigned to the unrelated pair, unlike the case of BERT.

Even by just looking at the different pairs in the table it is clear how topic segmentation fundamentally differs from entailing and paraphrasing sentences. While the paraphrases are clearly about a set of specific entities, in the case of the topic segmentation example, the relation between the two sentences mostly concern a semantic field that we could loosely refer to as *tallness*. The encoder fine-tuned on topic segmentation, then, as well as BERT seem to encode as important certain collocational relationships (*e.g.*, high, sky), whereas encoders fine-tuned just on paraphrases or entailment detection might have forgotten this kind of relationships as part of their specific fine-tuning.

At the same time, it looks like fine-tuning an encoder on topic segmentation as we did with TopSeg also allows it to recognise totally unrelated sentences in a more convincing way than using a general pre-trained language model such as BERT, suggesting that topic segmentation itself could be a valid pre-training strategy for text similarity-related tasks.

Overall, this section does not aim to give an exhaustive analysis, but these initial observations together with the analysis carried out so far are a strong indication that topic segmentation can rely on text similarity, but with a fundamentally different level of

granularity such that approaches otherwise effective for text similarity fail when encoding topically related sentences closer in space.

Future research can then draw on these initial conclusions in a variety of ways, such as exploring topic segmentation as a pre-training objective for general text similarity or adding the task to existing benchmarks in order to account for such differences in semantic granularity required.

## CONCLUSION

In this work we have evaluated the influence of pre-trained NSEs on topic segmentation. In doing so, our main contributions have been:

1) We established the superiority of models built on NSEs by drawing a comparison of such models against existing baselines across different datasets.
2) We presented a systematic comparison of NSEs for the task under a common framework including one supervised and one unsupervised model. As a result, we provided a ranking of different NSEs in the context of topic segmentation.
3) We have shown our results on three datasets representing three typical domains in which topic segmentation is used. By doing so, we have highlighted the differences between such domains in terms of topic segmentation performance of different models and NSEs, but also in terms of what the models might focus on in different situations.
4) We also introduced a metric, ARP, to directly evaluate the internal coherence of ground truth segments under different NSEs.

Our experiments pointed towards the following conclusions with respect to the use of NSE for topic segmentation:

1) Using NSEs as feature extractors mostly outperforms similar methods based on hierarchical networks, lexical or topical features.
2) The ranking of NSEs for the task does conform to existing literature on text similarity in certain respects (transformer-based models outperform other methods), but seems to diverge from it in others. Especially, models fine-tuned for text similarity and even for topic segmentation do not perform convincingly better than their non fine-tuned counterparts. Also, better pooling strategies failed to yield consistent improvements.
3) Following the above conclusion, we have shown that NSEs that perform better for topic segmentation do not always embed sentences from the same segment closer in space as opposed to sentences spanning multiple segments, as might be expected under the assumption that the models capture the underlying topical coherence of the data. Especially for domains like news shows and especially for supervised models that can learn what to focus on, this discrepancy is a strong hint towards the fact that topic segmentation does not always involve recognising the underlying topical coherence. We hypothesise that in these cases, the recognition of recurrent expressions or even individual words might be more relevant for good topic segmentation results.
4) Notwithstanding the above, our measure of relative topical coherence also highlighted how NSEs do a better job in encoding sentences from the same segment closer in space

when applied on natively textual data like Wikipedia articles, while performance drops with news shows transcripts, and their capacity to convincingly reduce inter-segmental embedding variance for meeting transcripts seems practically null. This is probably a result of the high frequency in this latter domain of utterances that may perform important dialogue functions but do not contain topical semantic content, and it is *per se* an interesting evidence of the limitations of well-established sentence encoders in such domains.

5) When text similarity plays a role in recognising topic boundaries, it looks like the type of text similarity involved in such a task is fundamentally different and probably looser than the one accounted for in traditional benchmarks for semantic similarity. This can be inferred by the poor performance both in topic segmentation and ARP scores shown by NSEs that performed very well on traditional sentence similarity benchmarks and it is further confirmed by qualitative analysis. This last piece of evidence, then, frames the task of topic segmentation as a fundamentally different task, such that a research direction worth pursuing is that of including this task in existing benchmarks for text similarity so as to account for the different levels of granularity at which we want to consider two pieces of text similar (*i.e.*, thematically/topically similar as opposed to be direct paraphrases or entailing each others).

Overall, our contribution is the first to our knowledge to compare different NSEs for topic segmentation in different domains. In doing so, we highlighted interesting convergences and divergences with existing literature and we have empirically shown how the task of topic segmentation might look very different in different domains.

In answering our third question, we introduced ARP scores as an indicator of the relative coherence of multi-topic documents under different NSEs, measuring the lexical cohesion of portions of text in the embedding space. This score is a relevant contribution by itself, as it can work as an effective feature selection method for topic segmentation, but it can also help answering a variety of other related questions.

Future research might investigate whether the use of ARP scores in conjunction with datasets such as Wikisection, in which the scores are highly correlated with topic segmentation performance, could make a good addition to existing benchmarks for evaluating NSEs performance. The task of topic segmentation, in fact, seems to have peculiar characteristics that make it different from usual text similarity tasks commonly used to evaluate NSEs; under this respect, adding the information of how these features work for topic segmentation could enrich existing benchmarks with a novel angle.

# APPENDICES

## Appendix A: importance of last sentence for supervised topic segmentation

From our experiments we could conclude that topic segmentation systems in different domains might focus on things different from lexical cohesion, especially where specific cues are present, which might be useful for segmentation.

**Table 7 New Correlation analysis between ARP Scores and the boundary similarity score obtained by re-training NSE-BiLSTM models on the three different corpora after having excluded the last sentence for each segment in every training set.**

| Dataset | Correlation coefficient | Difference with previous correlation |
|---|---|---|
| BBC3000 | 0.43 | +0.31 |
| QMSUM | 0.31 | −0.08 |
| Wikisection | 0.74 | +0.02 |

**Note:**
Pearson correlation coefficient has been used as correlation score.

This evidence has been indirectly observed by previous literature in, *e.g.*, dialogue topic segmentation, where cue sentences played a role in a number of proposed systems (*Joty, Carenini & Ng, 2013*). Similarly, in the media domain the use of specific cues have been largely used especially by early models, following the reasoning that in programmes such as news shows the anchor person tends to use formulaic expressions to move to a different topic (*Émilie & Quénot, 2012*; *Misra et al., 2010*; *Kannao & Guha, 2016*).

In this appendix to our main work, we further develop on our hypothesis for which lexical cohesion plays a minor role in automatic topic segmentation systems for domains in which specific segment-terminal cues are present. To do so, we re-run the NSE-BiLSTM experiments using the same settings and encoders we used in our main experiments, but this time we deleted the last sentence of each segment from the training set.

For every dataset, then, we applied a pre-processing step for which we delete all sentences or speaker turns $s_i^j$ with $j$ corresponding to any index where sentence $s_i^j$ is at the boundary between two topic segments in the training set document $S_i$. By doing so, we mask from the model any potential recurring cue phrase that the system could exploit to decide whether to predict a topic boundary.

Having applied this pre-processing step, we then re-train all the NSE-BiLSTM models following the same approach and parameters described in the main body of this work.

We then re-run the correlation analysis presented in "Correlation of ARP Scores with Topic Segmentation Results" to observe how the results from different encoders compare to ARP scores at this point.

The results from Table 7 clearly shows that for BBC3000 dataset the exclusion of the last sentence for each topic segment during training lead to the segmentation results for different encoders being much more correlated to our ARP scores for the same encoders. The correlation coefficient in fact is about 30 points higher and this evidence, together with the evidence previously collected, is a strong indicator towards what previously observed about this dataset. The big change in results, in fact, suggests that supervised models for this dataset were making a bigger use of localised cues from the last sentences of each segment, rather than modelling the underlying lexical cohesion of the entire segment.

If we look at Wikisection and QMSUM, instead, we can observe that the correlation coefficient does not vary significantly and this suggest that in these cases the effect of cue phrases was less pronounced.

### Funding

We received financial support from several sources: the Slovenian Research Agency *via* research core funding for the programme Knowledge Technologies (P2-0103), and the UK EPSRC *via* the projects Sodestream (Streamlining Social Decision Making for Improved Internet Standards, EP/S033564/1) and ARCIDUCA (Annotating Reference and Coreference In Dialogue Using Conversational Agents in games, EP/W001632/1). The funders had no role in study design, data collection and analysis, decision to publish, or preparation of the manuscript.

### Grant Disclosures

The following grant information was disclosed by the authors:
Slovenian Research Agency *via* research core funding for the programme Knowledge Technologies: P2-0103.
UK EPSRC *via* the projects Sodestream (Streamlining Social Decision Making for Improved Internet Standards): EP/S033564/1.
ARCIDUCA (Annotating Reference and Coreference In Dialogue Using Conversational Agents in games): EP/W001632/1.

### Competing Interests

Chris Newell is an employee of BBC R&D, London.

### Author Contributions

- Iacopo Ghinassi conceived and designed the experiments, performed the experiments, analyzed the data, performed the computation work, prepared figures and/or tables, authored or reviewed drafts of the article, and approved the final draft.
- Lin Wang conceived and designed the experiments, authored or reviewed drafts of the article, and approved the final draft.
- Chris Newell conceived and designed the experiments, authored or reviewed drafts of the article, and approved the final draft.
- Matthew Purver conceived and designed the experiments, authored or reviewed drafts of the article, and approved the final draft.

### Data Availability

The QMSUM dataset is described in more detail at Ming Zhong, Da Yin, Tao Yu, Ahmad Zaidi, Mutethia Mutuma, Rahul Jha, Ahmed Hassan Awadallah, Asli Celikyilmaz, Yang Liu, Xipeng Qiu, and Dragomir Radev. 2021. QMSum: A New Benchmark for Query-based Multi-domain Meeting Summarization. In Proceedings of the 2021 Conference of the North American Chapter of the Association for Computational Linguistics: Human Language Technologies, pages 5905–5921, Online. Association for Computational Linguistics.
The dataset is available at GitHub: https://github.com/Yale-LILY/QMSum.

The Wikisection dataset is described in more detail at Sebastian Arnold, Rudolf Schneider, Philippe Cudré-Mauroux, Felix A. Gers, Alexander Löser; SECTOR: A Neural Model for Coherent Topic Segmentation and Classification. Transactions of the Association for Computational Linguistics 2019; 7 169–184. https://doi.org/10.1162/tacl_a_00261.

The dataset is available at GitHub: https://github.com/sebastianarnold/WikiSection. Our code is available at GitHub and Zenodo:

- https://github.com/Ighina/NSE-TopicSegmentation.

- Iacopo Ghinassi. (2023). Ighina/NSE-TopicSegmentation: NSE-TopicSegmentation (v.1.0.0). Zenodo. https://doi.org/10.5281/zenodo.8245923.

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
