# Peer review of "Comparing neural sentence encoders for topic segmentation across domains: not your typical text similarity task"

_PeerJ Computer Science, doi:10.7717/peerj-cs.1593_

## Round 0.1 · original submission · Minor Revisions

The first round of refereeing is now complete. The three reviewers are largely positive about your submission. The paper structure is clear and the method is novel and interesting. Based on their feedback and my own reading of the manuscript I am pleased to accept your submission subject to addressing the minor concerns raised by the referees.

In particular, please pay attention to comments relating to basic reporting concerning the resolution for figures, grammatical errors and inconsistencies, and possible missing references. Please also clarify the choice of datasets for the experimentation.

**Language Note:** The review process has identified that the English language must be improved. PeerJ can provide language editing services - please contact us at copyediting@peerj.com for pricing (be sure to provide your manuscript number and title). Alternatively, you should make your own arrangements to improve the language quality and provide details in your response letter. – PeerJ Staff

·

Basic reporting

This paper explores Neural sentence encoders in the topic segmentation task. The paper is very large; its structure is clear, with many references.

Experimental design

The experiments included three datasets processed with different Transformer-based models like BERT and its modification. All NSE models showed good results.

Validity of the findings

The data is provided, and conclusions are well stated.

Additional comments

No additional comments

·

Basic reporting

I have read the paper entitled (Comparing neural sentence encoders for topic segmentation across domains: not your typical text similarity task). The manuscript is centered on an interesting topic. Organization of the paper is good and the proposed method is quite novel. Nevertheless, the manuscript does not link well with recent literature on Topic Segmentation appeared in relevant top-tier journals.

Though I found this paper well-written and there are only few grammatical errors which need to be corrected.
- Most sentences did not understand or they are grammatically incorrect.
- Proofreading is necessary for the manuscript for unabiguous sentences.

Equation numbers and its references not found. Table 2 placing is incorrect. Table 5 Placing is incorrect. Increse the resolution for Figure 2. In Figure 3 word - Non Boundary is half visible.

Experimental design

- Experimental design is satisfatory.
- Research question well defined, relevant & meaningful. It is stated how research fills an identified knowledge gap.
- Methods described with sufficient detail & information.

Validity of the findings

- Please explain why three datasets chosen for the experimentation to convince researches on the
efficiency of the proposed approach.
- The author did not present the contribution in detail. Justification to added contribution is required please.

Additional comments

Strength points of the manuscript
- The paper is good in terms of scientific structure, well written, language and the topic is
interesting to readers.
- They use the well know algorithms that are already did in many studies and recent also as bench mark.

Weakness points of the manuscript
- The author did not present the contribution in detail.
- The use of latest published references is neglected in this paper.

Reviewer 3 ·

Basic reporting

Figure 4 needs to be better resolution.

Missing Top2Vec reference on line 486.

There were a few inconsistent uses of Neural Sentence Encoders (NSEs), text encoders, neural NLP models from line 100-107.

Experimental design

Why is ARP better than other existing coherence measures?

Validity of the findings

Good description of datasets, but describe why those specific datasets were chosen.

---

## Round 0.2 · accepted · Accept

Thank you for submitting a revised manuscript. The resolution for figures is much improved and we appreciate the effort you've made to address the grammatical errors and inconsistencies. The paper now reads well. You've also satisfactorily clarified the choice of datasets for the experimentation. Based on feedback from the reviewer and my reading of the manuscript, I am happy to accept the revised manuscript for publication. Congratulations.

·

Basic reporting

No comment for improvements

Clear and unambiguous English used. Literature references, sufficient field background/context provided.

Experimental design

No comment for improvements

Research question well defined, relevant & meaningful. It is stated how research fills an identified knowledge gap.

Validity of the findings

No comment for improvements

Additional comments

Accept the research article.